# Image-Guided Ablation Therapies for Extrahepatic Metastases from Hepatocellular Carcinoma: A Review

**DOI:** 10.3390/cancers15143665

**Published:** 2023-07-18

**Authors:** Noriyuki Umakoshi, Yusuke Matsui, Koji Tomita, Mayu Uka, Takahiro Kawabata, Toshihiro Iguchi, Takao Hiraki

**Affiliations:** 1Department of Radiology, Okayama University Hospital, 2-5-1 Shikata-cho, Kitaku, Okayama 700-8558, Japan; kotomita@okayama-u.ac.jp (K.T.); mayuka@okayama-u.ac.jp (M.U.); t-kawabata@s.okayama-u.ac.jp (T.K.); 2Department of Radiology, Faculty of Medicine, Dentistry and Pharmaceutical Sciences, Okayama University, 2-5-1 Shikata-cho, Kitaku, Okayama 700-8558, Japan; y-matsui@okayama-u.ac.jp (Y.M.); hiraki-t@cc.okayama-u.ac.jp (T.H.); 3Department of Radiological Technology, Okayama University Graduate School of Health Science, 2-5-1 Shikata-cho, Kitaku, Okayama 700-8558, Japan; iguchi@ba2.so-net.ne.jp

**Keywords:** hepatocellular carcinoma, extrahepatic metastases, ablation therapy, radiofrequency ablation, microwave ablation, cryoablation, percutaneous ethanol injection

## Abstract

**Simple Summary:**

Although systemic therapies are a common treatment for patients with extrahepatic metastases from hepatocellular carcinoma (HCC), local ablative therapies, such as radiofrequency ablation, microwave ablation, and cryoablation, can be used in select patients who have metastases to the lung, bone, and other sites with curative or palliative intent. This review summarizes the currently available evidence on image-guided thermal ablation therapies for extrahepatic metastases from HCC.

**Abstract:**

The most common sites of extrahepatic metastases from hepatocellular carcinoma (HCC) are the lungs, intra-abdominal lymph nodes, bones, and adrenal glands, in that order. Although systemic therapies are a common treatment for patients with extrahepatic metastases, local ablative therapies for the extrahepatic metastatic lesions can be performed in selected patients. In this article, the literature on image-guided thermal ablation for metastasis to each organ was reviewed to summarize the current evidence. Radiofrequency ablation was the most commonly evaluated technique, and microwave ablation, cryoablation, and percutaneous ethanol injection were also utilized. The local control rate of thermal ablation therapy was relatively favorable, at approximately 70–90% in various organs. The survival outcomes varied among the studies, and several studies reported that the absence of viable intrahepatic lesions was associated with improved survival rates. Since only retrospective data from relatively small studies has been available thus far, more robust studies with prospective designs and larger cohorts are desired to prove the usefulness of thermal ablation for extrahepatic metastases from HCC.

## 1. Introduction

Hepatocellular carcinoma (HCC) is a primary liver tumor that often develops in patients with chronic liver disease, especially those with cirrhosis due to alcohol use, chronic hepatitis B or C viral infection, or non-alcoholic steatohepatitis [1,2]. According to the World Health Organization’s Global Cancer Observatory database, HCC is the fourth leading cause of cancer-related deaths worldwide, with extrahepatic metastases present in approximately 10–15% of cases at diagnosis in unsurveyed patients [3,4,5]. Extrahepatic metastases are more common in patients with advanced-stage primary tumors (>5 cm or macrovascular invasion) [4,6,7]. Patients with extrahepatic metastases have a poor prognosis, with a 5-year survival rate of only 3% [8,9]. The common sites of extrahepatic metastasis include the lungs, intra-abdominal lymph nodes, bones, and adrenal glands, and metastases to the pleura, peritoneum, and brain are also possible [10,11,12]. Extrahepatic metastases may also occur as a part of disease recurrence after local therapy for HCC in approximately 5–25% of patients [13,14,15].

In general, systemic therapies such as chemotherapy and molecularly targeted drugs are selected for extrahepatic metastases [16]. Alternatively, or in combination with systemic therapies, local therapies such as surgery, radiotherapy, and thermal ablation therapy can be performed with curative intent for oligometastases or with palliative intent for metastases causing pain [12,17,18]. Image-guided thermal ablation therapies, including radiofrequency ablation (RFA), microwave ablation (MWA), and cryoablation (CA), are preferred for patients who are ineligible for surgery due to the minimally invasive and repeatable characteristics of these therapies. Recent studies have shown the efficacy and safety of percutaneous ablative therapies for extrahepatic oligometastases in the lung, adrenal gland, bone, lymph node [LN], and pleura/peritoneum of HCC [19,20,21,22,23,24,25,26,27,28,29,30,31]. There is also a report regarding the efficacy of ablation in the treatment of painful bone metastases of HCC [28]. This review provides an overview of the current evidence regarding the outcomes of image-guided thermal ablation therapies for extrahepatic metastases of HCC.

## 2. Literature Search and Screening

PubMed was searched on 17 March 2023, using the following search terms: (“extrahepatic” [Title] OR “HCC” [Title] OR “hepatocellular carcinoma” [Title]) AND (“metastasis” [Title] OR “metastases” [Title] OR “metastatic” [Title] OR “oligometastases” [Title]) AND (“ablation” [Title] OR “ablative” [Title] OR “cryoablation” [Title] OR “laser interstitial thermal therapy” [Title] OR “LITT” [Title] OR “irreversible electroporation” [Title] OR “IRE” [Title]). Eighty-seven related articles were extracted, all published in 2003 or later.

The articles were screened using certain criteria based on their titles, abstracts, and texts. The criteria included: (i) articles written in English; (ii) clinical studies or meta-analyses; and (iii) studies showing outcomes, including survival after thermal ablation for extrahepatic metastases from HCC. When multiple studies conducted by the same institution had considerable overlap in terms of participants, only the study with the largest sample size was chosen.

Figure 1 shows the flowchart of the studies retrieved from the literature search and included in the analysis. Abstracts of the articles were reviewed according to pre-defined criteria. The full text of the subject papers was then evaluated, and data were extracted. An initial search identified 87 articles, of which 14 complete text articles were assessed and 11 were included [21,22,23,24,25,26,27,28,29,30,31].

## 3. Techniques of Thermal Ablation

Ablative therapies represent a focused approach to eliminating or significantly destroying localized tumors with reduced invasiveness compared to surgical removal. In the local treatment of HCC, RFA and MWA play a fundamental role due to their safety and efficacy in completely necrotic tumor nodes, with comparable complete response rates [32,33]. Percutaneous thermal ablation has also been established as an effective treatment for renal, lung, and bone tumors [34,35,36,37,38], as well as benign masses of the thyroid and uterus [39,40]. These therapies achieve irreversible destruction of tumor tissue by applying either hot or cold thermal energy.

RFA employs an applicator to administer high-frequency alternating current (at 400 and 500 kHz) to the targeted tissue, inducing ionic agitation and generating frictional heat (temperatures ranging from 60 to 100 °C). The extent of the ablation zone achieved depends on the impedance of the target tissue as well as the adjacent perfusion and ventilation.

MWA utilizes a 915-MHz or 2.45-GHz generator to produce an electrical current, which is delivered through a water-cooled interstitial antenna to create a localized, non-ionizing electromagnetic field. This field interacts with dipolar molecules, leading to frictional heating. MWA offers several theoretical advantages, such as a rapid temperature increase, a reduced heat-sink effect compared to RFA, and a larger ablation zone [41,42].

CA destroys tumors by subjecting them to extremely cold temperatures. This is accomplished by delivering argon gas at high pressure through a cryoprobe, causing rapid expansion and a sudden drop in temperature below 183 °C (known as the Joule-Thompson phenomenon). Consequently, intra- and extracellular water freeze, resulting in the disruption of cell membranes and organelle structures. Upon passive thawing, a fluid shift occurs from the interstitium into the tumor cells, leading to cell rupture. Intravascular ice crystals, direct endothelial freezing, microthrombi, and post-ablation edema contribute to indirect ischemia. The advantages of CA include preservation of the collagenous architecture and reduced intraprocedural pain [41,42].

Imaging techniques are employed for immediate preoperative planning, targeting, and intraoperative guidance. X-ray fluoroscopy, computed tomography (CT) (with or without CT fluoroscopy), and magnetic resonance imaging can be utilized for guidance either alone or in conjunction with 3D navigation or image fusion systems. Ultrasound guidance is appropriate when the tumor is clearly delineated in superficial regions or abdominal organs, allowing for the avoidance of adjacent sensitive structures.

The choice of ablation therapies and imaging modalities should be individualized based on the type of tumor, surrounding organs, and patient background.

## 4. Outcomes of Thermal Ablation for HCC Extrahepatic Metastases in Various Organs

Table 1 shows the results of studies on ablative therapies for metastases to various organs from HCC [21,22,23,24,25,26,27,28,29,30,31].

### 4.1. Lung

The lung is the most common site of extrahepatic metastases. Lung metastasis occurs in 20–39% of advanced-stage HCC [43,44]. The most common ablation technique for HCC lung metastases was RFA in previous studies (Table 1). The local efficacy exhibited a high degree of consistency and favorability throughout studies; tumors with a mean or median size of 14–19 mm were treated with local control rates of 83–92% [21,22,23,24,25] (Table 1). The reported 1-, 3-, and 5-year overall survival (OS) rates after lung ablation ranged from 73% to 89%, 26% to 70%, and 26% to 31%, respectively [21,22,23,24,25] (Table 1). Patients with up to two or three lung metastases showed better survival than those with more tumors [22,24]. The absence of viable or uncontrolled intrahepatic tumors was also associated with better survival [24,25]. Other factors were associated with a favorable prognosis, including a tumor diameter of less than or equal to 3 cm, lower serum α-fetoprotein (AFP) levels, Child-Pugh grade A, no liver cirrhosis, and no hepatitis C infection [24,25]. Yuan et al. [22] compared RFA, MWA, and CA for HCC lung metastases, finding no difference in survival outcomes among those different ablation methods. Wang et al. [21] investigated the outcomes of cone-beam CT-guided thermal ablation (RFA or MWA) and helical tomotherapy in 106 patients with pulmonary metastases from HCC. The 1- and 3-year OS rates were 75% and 26%, respectively, in the thermal ablation group (*n* = 63) and 77% and 37%, respectively, in the helical tomotherapy group (*n* = 43). The median OS was 18.0 months in the thermal ablation group and 23.4 months in the helical tomotherapy group (*p* = 0.38). These findings indicate that thermal ablation and helical tomotherapy exhibit comparable OS for pulmonary metastases from HCC. However, it should be noted that the ablation group had significantly higher Child-Pugh grades and AFP values compared to the helical tomotherapy group, which warrants caution in the interpretation of the results [21].

Major complications occurred in 2.5–25% of cases after ablation for HCC lung metastases, which were mostly pneumothorax requiring chest tube placement [21,22,23,24,25]. In addition, cases of pleural effusion requiring thoracocentesis drainage and massive hematomas caused by a pulmonary pseudoaneurysm have been reported [21,22,25].

### 4.2. LNs

The LNs represent the second most prevalent location for extrahepatic metastasis from HCC. During hepatectomy, LN metastases were detected in approximately 0.75% to 7.5% of HCC patients, as reported in previous studies [45,46], while autopsy findings indicated LN metastases in approximately 30.3% of patients [47]. The presence of metastases in the LNs significantly impacts the survival of affected individuals, with a median survival of merely three months observed in HCC patients with LN metastases in the absence of any therapeutic interventions [48]. A few studies are available on ablative therapies for LN metastases from HCC (Table 1) [26,27]. 

Yuan et al. [26] performed RFA, percutaneous ethanol injection (PEI), and MWA for LN metastases of HCC in 14, 5, and 12 patients, respectively, and showed that the 1-, 3-, and 5-year OS rates after LN ablation were 74.6%, 50.3%, and 50.3%, respectively. No significant differences were found in the accumulated OS rates among the three ablation techniques. The 1-, 3-, and 5-year rates of liver progression-free survival (LPFS) were 78.7%, 69.9%, and 69.9%, respectively. The local tumor control was reported to be 83.3% at 6 months. Pan et al. [27] documented that the 1-year OS rate was 58.3% after ablation for LN metastases. The local tumor control rate was reported to be 84.4% at 3 months after RFA. They retrospectively compared 46 matched pairs of patients undergoing RFA for LN metastases or not, using propensity score matching analysis, and found that RFA for LN metastases resulted in a 5.2-month median survival benefit. Their multivariate analyses showed that the presence of intrahepatic tumors (hazard ratio [HR] = 3.146, *p* = 0.001), the response to RFA (HR = 2.248, *p* = 0.017), and the number of LNs (HR = 2.013, *p* = 0.034) correlated with the OS after RFA.

In these two studies, major complications occurred in only one patient, who developed massive pleural effusion and severe pneumonia, requiring chest tube placement and anti-infection treatment [26].

### 4.3. Bone

The incidence of bone metastases in patients with HCC was 6–20% in autopsy studies [7,49] and 4–13% in clinical studies [50,51]. The median survival time of patients with bone metastases was 2.9 months without treatment, 5 months after cementoplasty, or 6 months after external beam radiotherapy [52,53]. Bone metastasis is often accompanied by pain and lowers the patient’s quality of life. Radiation therapy is commonly used for pain relief; however, it takes 3–4 weeks for symptom relief with radiotherapy, with pain elimination rates of 23–35% [54,55]. Only one retrospective study on ablative therapy for bone metastases from HCC was found (Table 1) [28]. Kashima et al. [28] reported the therapeutic results of RFA for 40 patients with 54 bone metastases, including 29 patients for whom pain relief was the purpose of RFA. The visual analog scale score was reduced by two points or more 1 week after bone RFA in 28 out of 29 patients (96.6%). A significant decrease was found in the mean visual analog scale score from 6.1 ± 2.5 (standard deviation) to 1.8 ± 1.7 (*p* < 0.001) after RFA. The 1-, 2-, and 3-year OS rates were 34.2%, 19.9%, and 10.0%, respectively, with a median survival time of 7.1 months. Factors significantly associated with a more favorable prognosis included complete ablation of bone metastases, the presence of a single bone lesion, negative AFP levels, and the absence of viable intrahepatic lesions. There were no major complications, except for a transient nerve injury observed in one patient (2.5%, 1/40), who experienced leg paralysis and vesicorectal dysfunction one day after undergoing RFA for bone metastasis in the fifth lumbar spinous process [28].

### 4.4. Adrenal Glands

Adrenal metastases from HCC are relatively rare, occurring in less than 10% of patients with HCC [11]. The prognosis for patients with HCC with adrenal metastases without effective treatment was poor [3]. Four reports have described the results of ablation therapy for adrenal metastases from HCC [29,30,31] (Table 1). Two retrospective studies have reported local tumor control of 74–79%, median OS of 14–16.8 months, and 1- and 2-year OS rates of 53–67% and 33%, respectively, in patients treated with RFA or MWA (Table 1) [29,30]. Lyu et al. [30] reported that 18 patients with adrenal oligometastases survived for a longer period than the nine patients with extra-adrenal metastases (21.8 months vs. 12.8 months, *p* = 0.037; HR, 0.330; 95% CI, 0.117–0.933). Technical success—defined as the disappearance of tumor enhancement on contrast-enhanced images 3–4 weeks after RFA—was 86.4–93.1% [29,30]. A tumor diameter of >3 or 5 cm was associated with an increased risk of residual tumors [29,30]. 

Yuan et al. [31] have reported that complete ablation of adrenal metastases with a mean size of 3.3 cm in diameter can be achieved with a combination of RFA and transarterial chemoembolization (TACE). Tumor-feeding arteries were embolized in TACE, which might reduce the cooling effect caused by blood flow (i.e., the heat sink effect) during RFA. They retrospectively compared treatment outcomes between patients who received RFA combined with TACE and TACE alone for adrenal metastasis. The 1-, 2-, and 3-year OS rates were 92.1%, 73.7%, and 55.3%, respectively, in the TACE+ RFA group and 88.0%, 64.0%, and 44.0%, respectively, in the TACE group. The RFA group combined with the TACE group exhibited higher survival rates than the TACE group. The mean survival time was 26.8 ± 2.0 months in the TACE + RFA group and 17.5 ± 2.2 months in the TACE group [31].

In ablation therapy for adrenal metastases from HCC, intraprocedural hypertension (>180 mmHg) occurred in 9–24.2% of cases [29,30,31]. The incidence of hypertension was higher in patients aged >65 years than in younger patients (66.7% [4/6 patients] vs. 14.8% [4/27 patients], *p* = 0.031) [30]. Huang et al. [29] reported myocardial transient ischemia occurring after RFA for adrenal metastasis in one patient (1/22, 4.5%).

## 5. Description in Guidelines

According to the Cardiovascular and Interventional Radiological Society of Europe (CIRSE) Standards of Practice on Thermal Ablation of Primary and Secondary Lung Tumours, ablation therapy is described as a treatment with level 2 evidence for colorectal lung metastases but with low evidence for pulmonary metastasis from other diseases, including HCC, lung cancers, renal cell carcinoma, melanoma, and sarcoma [41]. 

The CIRSE Standards of Practice for Thermal Ablation of Bone Tumors outline the indications for curative ablation therapy in selected patients with oligometastatic, oligorecurrent, and oligoprogressive disease without specifying the type of primary tumor [56]. Specifically, this therapy is recommended for the treatment of 3–5 potentially treatable metastases, each measuring less than 3 cm [57]. Ablation can be performed to prevent the compromise of critical structures adjacent to the tumor, particularly in spinal lesions [58]. Ablation can also be provided with palliative intent to alleviate painful metastases that are either unresponsive to or unsuitable for pharmacological management, radiation therapy, or surgery [59,60,61,62,63].

International guidelines for the treatment of HCC include the European Association for the Study of the Liver (EASL) Clinical Practice Guidelines [16] and the American Association for the Study of Liver Diseases (AASLD) Guidelines [64], but the indication for thermal ablation for extrahepatic metastases is not defined in these leading guidelines due to a lack of high-level evidence. The EASL Clinical Practice Guidelines on the Management of HCC recommend systemic therapy for the treatment of patients with extrahepatic metastases [16]. Although the EASL Guidelines refer to palliative radiotherapy for bone metastases causing pain or at significant risk of spontaneous secondary fracture, they do not mention local therapies such as surgery, radiotherapy, or thermal ablation therapy for other extrahepatic metastases.

## 6. Conclusions

Although the conventional approach for patients presenting with HCC extrahepatic metastases involves systemic therapy, image-guided thermal ablation therapy can be utilized for extrahepatic metastases, specifically affecting the lung, LNs, bone, and adrenal glands. RFA is considered to be the predominant technique, although MWA, CA, and PEI may also be employed. While the indication for ablation therapy for extrahepatic metastases has not been clearly defined, studies focusing on curative ablation for oligometastases have demonstrated the feasibility and safety of this treatment in the lungs, LNs, bone, and adrenal glands. Although the data are available from only retrospective studies, the survival outcomes of HCC patients after thermal ablation for extrahepatic metastases seem promising, considering the reported low survival rates of patients with stage 4 HCCs [8,9]. Sincee several studies indicated that the absence of viable intrahepatic lesions was associated with better survival, patients with a limited number of metastatic tumors and without intrahepatic lesions may be favorable candidates for curative thermal ablation of extrahepatic metastases. Additionally, ablation therapy for bone metastases may serve as a palliative intervention primarily aimed at pain alleviation. Further investigations involving larger cohorts and a prospective design are desired to validate the efficacy of ablation therapies for extrahepatic metastases from HCC. It is also warranted to explore the effectiveness of combining ablation therapy with systemic therapy and to compare its outcomes with other local therapies such as surgery and radiation therapy.

## Figures and Tables

**Figure 1 cancers-15-03665-f001:**
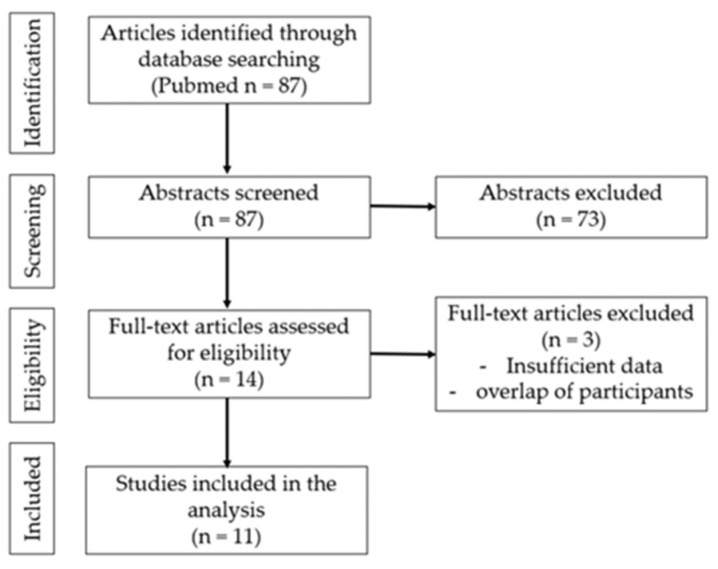
Flow chart of the search strategy.

**Table 1 cancers-15-03665-t001:** Outcomes of ablation for extrahepatic metastases from HCC.

Author, Year	Metastatic Site	Ablation Modality	Patients, *n* (Child–Pugh Grade A/B or C)	Tumors, *n*	Tumor Size [mm]	Follow-Up Period [mo]	Local Efficacy [%]	Survival	Major Complications
Wang et al., 2022 [21]	Lung	RFA or MWA	63 (52/11)	-	17.5	21.6 (mean)	-	1-/3-year OS: 75%/26%, Median OS: 18.0 mo	pneumothorax requiring CTP (3.2%), pleural effusion requiring drainage (1.6%)
Yuan et al., 2020 [22]	Lung	RFA or MWA or CA	39	-	15 (median)	13.5 (median)	84.2	1-/3-/5-year OS: 79.8%/58%/30.9%	pneumothorax requiring CTP (7.7%), pleural effusion requiring drainage (5.1%)
Lassandro et al., 2020 [23]	Lung	RFA	26	42	14 (mean)	-	-	1-/2-/3-/4-/5-year OS: 88.5%/69.8%/69.8% 34.9%/26.2%	pneumothorax requiring CTP (2.5%)
Li et al., 2012 [24]	Lung	RFA	29	68	19.3 (mean)	23 (median)	82.8	1-/2-/3-year OS: 73.4%/41.1%/30%, Median OS: 21 mo, 1-/2-year PFS: 59.7%/28.2%	pneumothorax requiring CTP (8.9%)
Hiraki et al., 2011 [25]	Lung	RFA	32 (27/5)	83	14 (mean)	20.5 (median)	92	1-/2-/3-year OS: 87%/57%/57%, Median OS: 37.7 mo	pneumothorax requiring CTP (23.1%), massive hematoma caused by a pulmonary pseudoaneurysm (1.5%)
Yuan et al., 2019 [26]	Lymph node	RFA or MWA or PEI	31 (31/0)	-	30 (median)	-	83.3	1-/2-/3-/4-/5-year OS: 74.6%/50.3%/50.3%/50.3%/50.3%, 1-/2-/3-year PFS: 24.7%/13.2%/0%	massive pleural effusion and severe pneumonia (3.2%)
Pan et al., 2017 [27]	Lymph node	RFA	46 (44/2)	62	32.4 (mean)	14.0 (median)	84.4	1-year OS: 58.3%, Median OS: 13.0 mo	none
Kashima et al., 2010 [28]	Bone	RFA	40 (28/12)	54	48 (mean)	11.9 (mean)	-	1-/2-/3-year OS: 34.2%/19.9%/10.0%, Median OS: 7.1 mo	transient nerve injury (2.5%)
Huang et al., 2019 [29]	Adrenal gland	RFA	22 (19/3)	22	40	10 (median)	73.7	1-/2-year OS: 52.6%/32.9%, Median OS: 14 mo	intraprocedural hypertension (9%), myocardial transient ischemia (4.5%)
Lyu et al., 2019 [30]	Adrenal gland	RFA or MWA	27 (25/2)	29	35 (mean)	19.3 (median)	77.8	OS 1-/2-year: 66.7%/33.3%, Median OS: 16.8 mo	intraprocedural hypertension (24.2%)
Yuan et al., 2018 [31]	Adrenal gland	RFA + TACE	38 (30/8)	38	33 (mean)	-	92.1	1-/2-/3-year OS: 92.1%/73.7%/55.3%, Mean OS: 26.8 mo	intraprocedural hypertension (15.8%)

HCC, hepatocellular carcinoma; RFA, radiofrequency ablation; MWA, microwave ablation; CA, cryoablation; PEI, percutaneous ethanol injection; TACE, transarterial chemoembolization; OS, overall survival; PFS, progression-free survival; CTP, chest tube placement.

## Data Availability

Data sharing not applicable.

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
