# Peer review of "Image-Guided Ablation Therapies for Extrahepatic Metastases from Hepatocellular Carcinoma: A Review"

_cancers, 2023, doi:10.3390/cancers15143665_

Round 1
Reviewer 1 Report
The review is nice and well written. I recommend to add some figures to improve the quality of the paper.
English grammar should be improved and revised by a native speaker.
I recommend to comment more on the state of the art of ablative treatments for HCC, citing the recent paper PMID: 33339274
Minor grammar revision
Author Response
The review is nice and well written. I recommend to add some figures to improve the quality of the paper.
Response: Thank you for your comment. We followed your suggestion and added a flow chart of the search strategy (Figure 1).
English grammar should be improved and revised by a native speaker.
Response: Thank you for your advice. We asked Editage (www.editage.com) to proofread the English again.
I recommend to comment more on the state of the art of ablative treatments for HCC, citing the recent paper PMID: 33339274
Response: Thank you for your advice. We added a section on ablative treatments for HCC in the Techniques in the thermal ablation section and added citations to the references you provided.
Reviewer 2 Report
This is a literature review of an interesting topic on local ablation techniques for HCC metastases. The article is reasonably well written and concise and the conclusions are appropriately tempered
Author Response
This is a literature review of an interesting topic on local ablation techniques for HCC metastases. The article is reasonably well written and concise and the conclusions are appropriately tempered.
Response: Thank you for your thoughtful review of our manuscript. We are very honored to receive your commendation for this review.
Reviewer 3 Report
Manuscript title: Image-guided ablation therapies for extrahepatic metastases
from hepatocellular carcinoma
Manuscript ID: cancers-2476333, Type:
Journal: Cancers
Article type: Review
The current review manuscript by Umakosh et al. “Image-guided ablation therapies for extrahepatic metastases from hepatocellular carcinoma” showed immense potential and outstanding match for publishing in Cancers Journal. I found the review article is up to the mark and highly recommend it for publication without any further revisions.
Author Response
The current review manuscript by Umakoshi et al. “Image-guided ablation therapies for extrahepatic metastases from hepatocellular carcinoma” showed immense potential and outstanding match for publishing in Cancers Journal. I found the review article is up to the mark and highly recommend it for publication without any further revisions.
Response: Thank you for your thoughtful review of our manuscript. We are very honored to receive your commendation for this review.
Reviewer 4 Report
I read with interest this study by Umakoshi et al., entitled ‘Image-guided ablation therapies for extrahepatic metastasis from hepatocellular carcinoma: a review’. This is a well written review and the literature search was well addressed. In fact, this topic has not been extensively explored, therefore international guidelines provided no strong recommendation addressing the best therapeutic options for extrahepatic metastasis.
I have only minor comments for the Authors.
1. A flow-chart providing information on the criteria for the selection or exclusion of articles starting from the initial number of 87 studies should be provided for readers’ better understanding .
2. In Table 1 adrenal grand should be changed in adrenal gland around the table
3. In the same Table, page 5 data by Kado et al., should be deleted since were not reported the years of survival but only median survival (so the information is not in line with that provided by other studies in Table 1).
4. The Table 1 layout could be ameliorated ..for example in page 4 the line spacing was not invariably the same and also space between words.
5. The sentence in page 5, lines 141-142 should be better explained, since it is true that the OS was similar in participants who underwent ablation or helical tomotherapy, but the ablation group had more advanced liver disease and more aggressive tumor ( >CP and > AFP).
6. Data on peritoneum and pleura are very few and not well explained, therefore I suggest to delete this section or better describe the results.
7. Page 7, line 238 change select in selected.
8. Finally, ‘information or no information’ from other international guidelines over than EASL should be provided on the topic of therapies for extrahepatic metastasis from HCC.
Author Response
I read with interest this study by Umakoshi et al., entitled ‘Image-guided ablation therapies for extrahepatic metastasis from hepatocellular carcinoma: a review’. This is a well written review and the literature search was well addressed. In fact, this topic has not been extensively explored, therefore international guidelines provided no strong recommendation addressing the best therapeutic options for extrahepatic metastasis.
I have only minor comments for the Authors.
Response: Thank you for your thoughtful review of our manuscript. We appreciate your insightful comments, which have helped us improve our paper. We have revised the manuscript based on your suggestions. Please find our point-by-point responses to your comments below.
- A flow-chart providing information on the criteria for the selection or exclusion of articles starting from the initial number of 87 studies should be provided for readers’ better understanding .
Response: Thank you for your comment. We followed your suggestion and added a flow chart of the search strategy (Figure 1).
- In Table 1 adrenal grand should be changed in adrenal gland around the table
Response: Thank you for pointing out the spelling mistake. We revised it to “adrenal gland”.
- In the same Table, page 5 data by Yamakado et al., should be deleted since were not reported the years of survival but only median survival (so the information is not in line with that provided by other studies in Table 1).
Response: Thank you for your kind advice. Your point is correct. We deleted the results of Yamakado et al. from Table 1 according to your advice.
- The Table 1 layout could be ameliorated ..for example in page 4 the line spacing was not invariably the same and also space between words.
Response: Thank you for your kind advice. Table 1 is presented in a horizontal format, and displayed on a single page for better presentation.
- The sentence in page 5, lines 141-142 should be better explained, since it is true that the OS was similar in participants who underwent ablation or helical tomotherapy, but the ablation group had more advanced liver disease and more aggressive tumor ( >CP and > AFP).
Response: Thank you for your kind advice. In response to your suggestion, we revised that section as follows.
“Wan et al. [21] investigated the outcomes of thermal ablation (RFA or MWA) and helical tomotherapy in 106 patients with pulmonary metastases from HCC. The 1- and 3-year OS rates were 75% and 26%, respectively, in the thermal ablation group (n=63) and 77% and 37%, respectively, in the helical tomotherapy group (n=43). The median OS was 18.0 months in the thermal ablation group and 23.4 months in the helical tomotherapy group (P=0.38). These findings indicate that thermal ablation and helical tomotherapy exhibit comparable OS for pulmonary metastases from HCC. However, it should be noted that the ablation group had significantly higher Child-Pugh grade and AFP values compared to the helical tomotherapy group, which warrants caution in the interpretation of the results [21].”
- Data on peritoneum and pleura are very few and not well explained, therefore I suggest to delete this section or better describe the results.
Response: Thank you for your kind advice. Per your suggestion, we have removed section 4.5. Peritoneum and pleura.
- Page 7, line 238 change select in selected.
Response: Thank you for your advice. We have made the revision.
- Finally, ‘information or no information’ from other international guidelines over than EASL should be provided on the topic of therapies for extrahepatic metastasis from HCC.
Response: Thank you for your advice. We mentioned the EASL Guidelines as well as the American Association for the Study of Liver Diseases (AASLD) Guidelines as international guidelines for the treatment of HCC. We also noted that there is no indication of thermal ablation for extrahepatic metastases in these leading guidelines.
Round 2
Reviewer 1 Report
The revised version of the manuscript is OK. Thank you!
Author Response
Thank you for your thoughtful review of our manuscript. We are very honored to receive your commendation for this review.